# Oral Microbiome Dysbiosis as a Risk Factor for Stroke: A Comprehensive Review

**DOI:** 10.3390/microorganisms12081732

**Published:** 2024-08-22

**Authors:** Georgy Leonov, Diana Salikhova, Antonina Starodubova, Andrey Vasilyev, Oleg Makhnach, Timur Fatkhudinov, Dmitry Goldshtein

**Affiliations:** 1Federal Research Center of Nutrition, Biotechnology and Food Safety, 109240 Moscow, Russia; avs.ion@yandex.ru; 2Institute of Molecular and Cellular Medicine, RUDN University, 117198 Moscow, Russia; diana_salikhova@bk.ru (D.S.); vav-stom@yandex.ru (A.V.); tfat@yandex.ru (T.F.); 3Research Centre for Medical Genetics, 115522 Moscow, Russia; buben6@yandex.ru (O.M.); dvgoldrm7@gmail.com (D.G.); 4Therapy Faculty, Pirogov Russian National Research Medical University, 117997 Moscow, Russia; 5E.V. Borovsky Institute of Dentistry, I.M. Sechenov First Moscow State Medical University of the Ministry of Health of the Russian Federation (Sechenov University), 119991 Moscow, Russia; 6Central Research Institute of Dental and Maxillofacial Surgery, 119021 Moscow, Russia

**Keywords:** stroke, oral microbiome, dysbiosis, gut microbiome, oral health, periodontitis, dental caries, atherosclerosis, endothelial dysfunction

## Abstract

Stroke represents a significant global health burden, with a substantial impact on mortality, morbidity, and long-term disability. The examination of stroke biomarkers, particularly the oral microbiome, offers a promising avenue for advancing our understanding of the factors that contribute to stroke risk and for developing strategies to mitigate that risk. This review highlights the significant correlations between oral diseases, such as periodontitis and caries, and the onset of stroke. Periodontal pathogens within the oral microbiome have been identified as a contributing factor in the exacerbation of risk factors for stroke, including obesity, dyslipidemia, atherosclerosis, hypertension, and endothelial dysfunction. The alteration of the oral microbiome may contribute to these conditions, emphasizing the vital role of oral health in the prevention of cardiovascular disease. The integration of dental and medical health practices represents a promising avenue for enhancing stroke prevention efforts and improving patient outcomes.

## 1. Introduction

Stroke is one of the leading causes of death, morbidity, and long-term disability worldwide. Stroke survivors frequently experience complications such as physical or cognitive impairments and speech difficulties. Approximately 30–50% of stroke survivors suffer from severe disability [1]. Stroke is a neurological disorder resulting from a partial or complete lack of blood supply to any part of the brain, often because of impaired blood flow (ischemic stroke, IS) or leakage from a ruptured brain vessel (hemorrhagic stroke, which may be further subdivided into intracerebral hemorrhage (ICH) and subarachnoid hemorrhage (SAH)) [2,3]. The pathophysiology of stroke is a highly complex phenomenon, encompassing a multitude of cascading processes. These include loss of cellular homeostasis, energy deficiency, metabolic acidosis, elevated intracellular Ca^2+^ levels, free radical-mediated toxicity, cytokine-mediated cytotoxicity, apoptosis, autophagy, and disruption of the blood–brain barrier (BBB) [4].

The prevalence of behavioral and metabolic risk factors for cardiovascular disease (CVD)—such as smoking, alcohol consumption, high sodium diet, high systolic blood pressure (SBP), high low-density lipoprotein (LDL) cholesterol, renal dysfunction, high fasting plasma glucose (FPG), and high body mass index (BMI)—has increased with rapid economic development, changes in social ideology, and population aging [5,6]. Furthermore, additional risk factors for hemorrhagic stroke include aneurysms, arteriovenous malformations, and brain injury. Early diagnosis and reduction of stroke risk factors are the most effective approaches to preventing mortality and disability and minimizing the burden on healthcare systems worldwide [7].

Identifying new biomarkers associated with stroke is a promising research direction. The human body is host to a vast and diverse community of microorganisms, collectively termed the microbiome. These microbial populations, which inhabit a variety of niches, including the gut, skin, mouth, and urogenital tract, play a pivotal role in maintaining homeostasis and influencing a range of physiological processes [8]. Advances in metagenomics and high-throughput sequencing technologies have significantly expanded our understanding of the microbiome’s composition and function, revealing intricate interactions between host and microbes that can impact health and disease [9]. Over the past decade, many studies have shown an association between the host microbiome and stroke Over the past decade, many studies have shown an association between the host microbiome and stroke, both in terms of likelihood of development and stroke outcomes [10,11]. For instance, there is evidence that the gut microbiome plays a role in the modulation of systemic inflammation, which is a key factor in the pathogenesis of stroke [12]. Consequently, an investigation of the microbiome as a potential biomarker for stroke could yield valuable insights into disease mechanisms and facilitate the development of novel therapeutic interventions (Figure 1).

The oral microbiome is a highly diverse community of microorganisms inhabiting the human oral cavity [13]. It comprises bacteria, archaea, fungi, protozoa, and viruses, with an estimated total number of species exceeding 700 [14]. The human oral microbiome predominantly consists of members of the phyla *Actinobacteria*, *Proteobacteria*, *Firmicutes*, *Bacteroidetes*, and *Fusobacteria*. The five major genera identified in the oral cavity are *Streptococcus*, *Prevotella*, *Veillonella*, *Neisseria*, and *Haemophilus* [15,16]. According to the expanded Human Oral Microbiome Database (HOMD), 58% of species have been assigned formal names, 16% are unnamed but cultivated, and 26% are known solely as uncultivated phylotypes [17]. The oral microbiome has been demonstrated to exhibit individuality and relative stability in healthy individuals [18]. However, the composition and proportion of microorganisms may be influenced or altered by various factors, including heredity, dietary habits, smoking, alcohol consumption, oral diseases, socioeconomic status, antimicrobial use, and pregnancy [19,20,21,22,23,24,25,26]. Dysbiosis, defined as an alteration in the composition and function of one or more microbial communities, may be characterized by a loss or gain of specific microorganisms or by a loss of overall microbial diversity [27].

The evidence suggests that smoking can result in substantial alterations to the composition of the oral microbiome. The study demonstrated that the relative abundance of *Neisseria* and *Capnocytophaga* was reduced, while *Streptococcus* and *Megasphaera* were increased in current smokers [28]. Additionally, numerous microbial characteristics associated with cigarette smoking were identified as partial mediators of the associations between cigarette smoking and serum triglyceride and C-reactive protein levels [29]. Furthermore, alcohol consumption has a pronounced effect on the oral microbiota. Therefore, the abundance of *Actinomyces*, *Leptotrichia*, *Cardiobacterium*, and *Neisseria* was higher, while *Lactobacillales* was lower in individuals with high alcohol consumption [30]. It is evident that socioeconomic status plays a pivotal role, as it largely determines the availability of medical care, including dental care. This, in turn, increases the likelihood of developing and progressing oral diseases.

The most prevalent oral diseases, including dental caries, gingivitis, and periodontitis, are caused by the activity of microorganisms [31,32]. Periodontal disease represents a significant public health concern, affecting an estimated 20–50% of the global population [33]. The prevalence of dental caries in the local community varies widely, ranging from 25% to 99%, and root caries from 8% to 74%, depending on the population [34].

New evidence suggests a link between the oral microbiome and various neurological and mental diseases such as Alzheimer’s disease, Parkinson’s disease, multiple sclerosis, Lewy body disease, schizophrenia, and brain injury [35,36,37,38,39,40].

The objective of this review is to summarize the knowledge of the relationship between oral microbiome dysbiosis and stroke. It presents data on the association between microorganism-associated oral diseases and stroke development, as well as the mechanisms of bacterial influence on stroke risk factors.

## 2. Oral Diseases and Stroke

Periodontitis is a chronic inflammatory disease of multifactorial etiology. It is characterized by the progressive destruction of the dental apparatus, which results in the loss of periodontal supporting tissues and resorption of alveolar bone [41]. Specific genera, including *Porphyromonas*, *Treponema*, *Campylobacter*, *Eubacterium*, and *Tannerella*, have been identified at high levels in periodontitis sites, while other genera, such as *Veillonella*, *Neisseria*, *Rothia*, *Corynebacterium*, and *Actinomyces*, were highly prevalent in the healthy gingival sulcus [42].

The majority of studies have indicated a correlation between the prevalence of periodontitis and the occurrence of stroke. However, a number of studies have also demonstrated that this correlation is not universal or that it is specific to a certain subtype of stroke [43,44]. A meta-analysis of prospective and retrospective studies demonstrated that the adjusted risk of stroke in subjects with periodontitis was 1.47 times higher than that in subjects without periodontitis (95% confidence interval, 1.13–1.92; *p* = 0.0035) in prospective studies and 2.63 times higher in retrospective studies (95% confidence interval, 1.59–4.33; *p* = 0.0002) [45]. Moreover, a study of 6460 Americans over 30 years of age found that severe and moderate periodontitis was associated with a 2.55- and 1.71-times higher risk of stroke, respectively [46]. Another study using data from the U.K. Biobank datasets and the European MEGASTROKE consortium showed that chronic periodontitis was associated with cardioembolic stroke but not with ischemic stroke [47]. A study of young adults with cryptogenic ischemic stroke demonstrated that severe periodontitis (OR of 7.48 (1.24–44.9)) and invasive dental procedures (OR of 2.54 (1.01–6.39)) were associated with an elevated risk of developing cryptogenic stroke [48]. Nevertheless, a study conducted in a Swedish population found an association between periodontitis and mortality and coronary heart disease but no significant difference in the likelihood of stroke [44]. Studies in C57BL/6 mice have demonstrated that periodontitis is associated with increased chronic inflammation, particularly elevated levels of TNF-α and tissue infiltration of Th17 cells and IL-17 + γδ T cells. Nevertheless, these effects do not influence the outcome of stroke or the destruction of the blood–brain barrier [49].

Furthermore, observational studies have underscored the importance of dental health in the context of stroke. The correlation between periodontitis and stroke was found to be significantly higher than that between periodontitis and coronary artery disease [50]. Poor oral hygiene has been associated with an increased risk of developing metabolic disorders, which may in turn increase the risk of stroke [51]. Moreover, a 26-year cohort study indicated that gingival inflammation may also be an independent risk factor for stroke [52].

A study by Sen et al. examined the presence of dental caries in 6326 participants without previous stroke and found that subjects with dental caries exhibited a higher risk of ischemic stroke compared to subjects without caries. The association remained significant after controlling for potential confounding variables, including race, sex, age, education, body mass index, diabetes, smoking status, and the presence of periodontitis [53]. Another cohort study by Chang et al. also demonstrated a positive association between the number of dental caries (≥4) and the risk of stroke. Nevertheless, the relationship between dental caries and stroke remains a subject of contention [54]. Some studies have indicated that there is no correlation between dental caries and transient ischemic attack or acute ischemic stroke. A recent study conducted on a Korean population found that frequent toothbrushing (≥3 times per day) was negatively associated with the risk of stroke. The number of teeth affected by dental caries (≥4) was found to be positively associated with the incidence of stroke [55]. Furthermore, tooth loss has been identified as an independent risk factor for stroke. Individuals with severe tooth loss (≥23 missing teeth) exhibited a greater atherosclerotic burden and poorer stroke outcomes, as assessed by the modified Rankin Scale (mRS) [56]. Moreover, a study of 12,764 older men demonstrated that the loss of >10 teeth was an independent predictor of the risk of a cerebral infarction [57]. The most recent research findings in this field are presented in Table 1.

Three theories have been proposed to explain the mechanism of the connection between periodontitis and atherosclerotic plaques. The bacterial invasion theory proposes a direct effect of bacteria and their toxins on the endothelium. The cytokine theory emphasizes the role of inflammatory mediators released by cells of the immune system in damage to the endothelium of the vascular wall [58]. The autoimmunization theory emphasizes the importance of heat shock proteins (HSP65) expressed on oral pathogens such as *Porphyromonas gingivalis*, *Prevotella intermedia*, and *Actinobacillus actinomycetemcomitans* [59].

The systemic inflammatory response is elevated in patients with periodontitis in comparison to patients without periodontitis, indicating a robust correlation with diseases sensitive to the level of chronic inflammation. Bacterial lipopolysaccharides released into the bloodstream of patients with chronic infections such as periodontitis induce the production of acute phase proteins such as C-reactive protein (CRP) [60]. The concentration of CRP is consistently elevated in patients with periodontitis in comparison to healthy controls [61]. Acute phase proteins accumulate in damaged blood vessels, promoting the activation of phagocytes, which release nitric oxide, thereby facilitating the formation of atheromas. The chronic elevation of CRP levels in patients with periodontitis serves to exacerbate the inflammatory processes that occur within atherosclerotic plaques. It is generally accepted that inflamed plaques are unstable and prone to rupture, increasing the risk of cerebrovascular events [62]. Research has demonstrated that prolonged exposure to common infections increases the risk of stroke, while acute infections can act as stroke triggers. It is well established that inflammation, which is part of the body’s defense response to surgery or injury, plays an important role in the development of cardiovascular diseases such as heart attack or ischemic stroke [63]. Poor oral hygiene can result in the invasion of oral bacteria and the subsequent release of inflammatory mediators, which may increase the risk of stroke [54].

Oral diseases are associated with an elevated risk of stroke because of the systemic inflammatory and immune responses. Studies have shown that individuals with periodontitis and dental caries have a significantly higher risk of stroke than healthy individuals. Furthermore, poor oral hygiene and dental caries are also associated with an increased risk of cerebral infarction, underscoring the importance of oral health in stroke prevention.

**Table 1 microorganisms-12-01732-t001:** Recent studies investigating the association of oral disease with stroke.

Stroke or Associate Condition	Oral Disease	Year	Population	Brief Results	Reference
Stroke	Severe periodontitis and moderate periodontitis	2023	6460 Americans aged 30 years and older	Stroke was associated with the severity of periodontitis. The risk of stroke in participants with severe and moderate periodontitis was 2.55 and 1.71 times higher, respectively, than in participants without periodontitis.	Zheng et al. [46]
Ischemic, cardioembolic stroke, large artery atherosclerosis	Chronic periodontitis and aggressive periodontitis	2023	U.K. Biobank datasets (950 cases and 455,398 controls), European ancestry (851 cases and 6836 controls), European MEGASTROKE consortium (34,217 cases and 406,111 controls)	A correlation has been demonstrated between chronic periodontitis and cardioembolic stroke. However, no association was identified between periodontitis and ischemic stroke or large artery atherosclerosis.	Ma et al. [47]
Atrial fibrillation, stroke	Periodontitis	2021	5958 participants from Atherosclerosis Risk in Communities Study	Severe PD was associated with AF. Mediation analysis suggested that AF mediated the association between PD and stroke. Regular dental care was associated with a lower risk of developing AF.	Sen et al. [64]
Ischemic stroke	Periodontitis Administration Saliva samples from periodontitis patients	2022	C57BL/6 mice	Mice that were gavaged with periodontitis salivary microbiota exhibited significantly worse stroke outcomes and more severe neuroinflammation. Additionally, there was an accumulation of Th17 cells and IL-17 + γδ T cells in the ileum.	Chen et al. [49]
Stroke, myocardial Infarction	Periodontitis	2021	298,128 participants from the Korean National Health Insurance Screening Group	A significant association was observed between periodontitis and an increased risk of myocardial infarction (11%) and stroke (3.5%).	Cho et al. [65]
Stroke, coronary heart disease	Periodontitis	2021	1850 participants aged 47–73 years from the EGAT study	A significant association was found between severe periodontitis and the incidence of coronary artery disease compared with the group without periodontitis or the mild periodontitis group. However, there was no significant association when considering overall cardiovascular disease events and stroke outcomes.	Tiensripojamarn et al. [50]
Stroke, ischemic heart disease, mortality	Periodontitis	2021	1402 individuals over the age of 60 in the Swedish population	A correlation has been identified between periodontitis and the incidence and mortality of coronary heart disease; however, no similar correlation has been observed between periodontitis and stroke.	Bengtsson et al. [44]
Stroke	Periodontitis	2024	26,901 participants across 32 countries	A correlation has been demonstrated between tooth loss and stroke. However, the presence of painful teeth did not contribute to this association. The combination of all three symptoms, including tooth loss, painful teeth, and painful gums, strengthened the association with stroke compared to tooth loss alone.	Murphy et al. [66]
Stroke	Periodontitis	2019	C57BL/6 mice	The presence of periodontitis was associated with elevated levels of chronic inflammation and an augmented concentration of TNF-α. Nevertheless, no significant differences were observed in infarct volume or blood–brain barrier destruction.	O’Boyle et al. [67]
Ischemic stroke, hemorrhagic stroke, myocardial infarction	Periodontitis	2023	46,737 older adults from the Korea National Health Insurance Service—Senior Cohort Database	In the individual outcome analysis, ischemic stroke and myocardial infarction were associated with chronic periodontitis severity, but hemorrhagic stroke was not.	Jang et al. [68]
Minor ischemic stroke, transient ischemic attack	Periodontitis	2022	The study included 792,426 young adults with periodontitis and an equivalent number of controls	The risk of developing TIA and minor ischemic stroke was significantly elevated in participants with periodontitis (HR, 1.24; 95% CI, 1.15–1.32; *p* < 0.001) compared to the control group. The hazard ratio was marginally elevated among younger individuals (20–40 years).	Lee et al. [69]
Cryptogenic stroke	Periodontitis	2024	146 case–control pairs of young adults (cryptogenic ischemic stroke/control)	Cryptogenic ischemic stroke was found to be associated with high-grade periodontal inflammation with an OR of 10.48 (3.18–34.5) and severe periodontitis with an OR of 7.48 (1.24–44.9). The occurrence of invasive dental procedures within a three-month period prior to the onset of CIS was associated with an odds ratio of 2.54 (95% CI: 1.01–6.39).	Leskelä et al. [48]
Stroke	Dental caries, poor oral hygiene	2021	206,602 participants from the Korean National Health Insurance Screening Group	Frequent tooth brushing (≥3 times per day) was negatively associated with the risk of stroke. The number of teeth affected by dental caries (≥4) was found to be positively associated with the incidence of stroke.	Chang et al. [54]
Stroke	Dental caries	2023	6351 participants from Atherosclerosis Risk in Communities	The presence of ≥1 dental caries was associated with an elevated risk of stroke. An increase in the number of destroyed, missing, and filled surfaces was significantly associated with both stroke and death (from any cause) but not with coronary heart disease.	Sen et al. [53]
Stroke	Dental caries	2023	The study included 23,662 individuals from the Taiwan National Health Insurance Database	A stratified analysis revealed a positive association between advanced/severe dental caries and the risk of ischemic stroke in patients aged ≥40 years and hemorrhagic stroke in patients aged <40 years.	Ono et al. [70]
Stroke	Dental caries	2020	1742 stroke patients and 1193 healthy individuals	Stroke patients were found to have poorer oral health, with more dental caries, higher plaque index, and higher gingival index.	Zeng et al. [71]
Stroke	Tooth loss	2020	418 patients with ischemic stroke or transient ischemic attack	Atherosclerotic burden ≥50% and mRS scores were independently associated with severe tooth loss (>23 missing teeth).	Leao et al. [56]
Stroke, myocardial infarction, heart failure	Tooth loss	2019	4,440,970 participants included in the KNHI study	Tooth loss was an independent risk factor for cardiovascular events. Each missing tooth was associated with approximately a 1% increase in myocardial infarction, a 1.5% increase in heart failure and stroke, and a 2% increase in mortality. Having ≥5 missing teeth significantly increased the risk of cardiovascular outcomes.	Lee et al. [72]
Cerebral infarction	Tooth loss	2019	12,764 older men in the Norwegian population	The loss of more than 10 teeth was an independent predictor of the risk of a cerebral infarction.	Håheim et al. [57]

## 3. Oral Bacteria and Stroke

The study of the influence of microbial diversity and individual members of the oral microbiome on the development and outcome of stroke is of significant interest. Oral bacteria can have complex effects on the host, increasing systemic inflammation, penetrating the blood and brain, and releasing toxins. In addition, once they enter the intestines, they can change the structure of the intestinal microbiome [73].

Currently, there are few studies examining microbial diversity in stroke patients. The evidence regarding the impact of oral microbial diversity on stroke is conflicting. The findings of a study conducted on a sample of 146 individuals indicated a notable enhancement in the alpha diversity (the average number of species living in a particular niche observed in patients who had experienced a stroke) [74]. In contrast, another study among Chinese women demonstrated an inverse relationship between alpha diversity and stroke risk [75]. However, the underlying cause of this phenomenon remains unclear. One hypothesis suggests that this may be due to decreased mobility and, as a consequence, poorer oral hygiene [76]. Another study found that *Treponema pedis* and *Streptococcus suis* were present in a significant percentage of patients with stroke, whereas these bacteria are not normally found [77]. Additionally, there was an increase in the prevalence of *Pseudomonas aeruginosa*, *Klebsiella pneumoniae*, and *Escherichia coli* in the oral cavity one month following the stroke [78]. The abundance of *Corynebacterium*, *Lautropia*, and *Selenomonas* was found to be associated with an elevated risk of stroke [75]. Another study was conducted to investigate the oral microbiome in young individuals with cryptogenic stroke. The results indicated an increase in the abundance of *Brenneria goodwinii*, *Variovorax boronicumulans*, *Pseudomonas* sp. *AN-B15*, and *Actinoalloteichus* sp. *GBA129-24*, and a decrease in *Mycoplasma alkalescens*, *Streptomyces* sp. *LBUM 1475*, *Pseudonocardia dioxanivorans*, *Dokdonella*, and *Francisella frigiditurris* [79].

The oral cavity is home to two main types of pathogenic microorganisms: those that cause dental caries (cariogenic pathogens) and those that cause periodontitis (periodontal pathogens) [80,81]. *Streptococcus mutans*, which also belong to the early colonizers of the oral cavity, can be classified as a cariogenic pathogen [82]. Consequently, strains of *Streptococcus* expressing cell surface collagen-binding protein (Cnm) were found to be associated with an increased risk of cerebral microbleeds [83]. Furthermore, there is evidence indicating that the prevalence of streptococcal species within the tissue of patients’ blood clots is 5.1 times higher than in arterial blood [84]. Moreover, several studies have demonstrated a high prevalence of these bacteria in blood vessels. For instance, 78.9–84.8% of thrombi, 80.0% of carotid endarterectomy specimens, and 31.3–50% of carotid arteries were found to be colonized by these bacteria [85,86]. At the same time, it was shown that a reduction in the prevalence of *Streptococcus* in the oral cavity was a risk factor for stroke-associated pneumonia [87].

The majority of periodontal pathogens are Gram-negative, obligate anaerobic bacteria that inhabit periodontal pockets [88]. The most significant periodontal pathogens include *P. gingivalis*, *P. intermedia*, *T. denticola*, *A. actinomycetemcomitans*, *Tannerella forsythia*, and *Fusobacterium nucleatum* [89]. There is evidence for a higher abundance of *P. gingivalis* in stroke patients. *A. actinomycetemcomitans* was higher in patients with diabetic nephropathy and was also associated with a higher incidence of cerebral infarction. In addition, *P. gingivalis* LPS has been shown to promote Cdc42 activation of platelets and also impair blood clotting [90]. Moreover, *P. gingivalis* was found in 33.7% of thrombi in stroke patients and is also associated with a lower rate of complete reperfusion and negative stroke outcome [91]. In addition, the level of antibodies against *F. nucleatum* was negatively associated with stroke outcome [92]. A positive titer of serum antibodies to *Campylobacter rectus* has been associated with the presence of cerebral microbleeds in patients with acute stroke [93]. Several studies have found high percentages of *P. gingivalis*, *T. forsythia*, and *T. denticola* in atherosclerotic plaque tissue [94,95]. Remarkably, a recent study showed that levels of *P. gingivialis* and *T. forsythia* were lower in individuals with high levels of multimorbidity, including stroke [96]. Information on recent research in this area is presented in Table 2.

Despite the existence of several studies indicating a correlation between specific oral microbiome constituents and stroke occurrence, the underlying pathogenic mechanisms remain uncertain. While a number of species have been identified in both atherosclerotic plaques and blood clot tissue from stroke patients, further research is necessary to ascertain the specific alterations they may induce.

## 4. Link between Oral Microbiome and Stroke Risk Factors

### 4.1. Chronic Inflammation

Systemic inflammation is currently considered one of the key theories explaining the impact of oral dysbiosis on other organs and tissues, as well as its negative contribution to the development of many systemic diseases [100,101,102]. Recognition of periodontal pathogens by the body results in increased clonal expansion and abundance of immune cells such as monocytes, B and T lymphocytes, and dendritic cells in oral tissues [103]. Monocyte chemoattractant protein-1 (MCP-1) played an important role in this process, which promotes the activation and recruitment of immune cells into periodontal pockets [104]. In addition, damage to gingival epithelial cells can lead to additional recruitment of neutrophils through activation of chemokine receptor 2 (CXCR2) [105]. Cytokines are secreted by mononuclear phagocytes, antigen-presenting cells, lymphocytes, and neutrophils. The most studied proinflammatory cytokines and tissue-damaging agents in the circulatory system, such as IL-1β, IL-10, IL-17, Th17, IFN-γ, GM-CSF, G-CSF, IL-8, TNF-α, and MCP1, are released during the oral tissue response after pathogen invasion [106]. Fibroblasts, osteoclasts, and osteoblasts are able to attract immune cells through activation of the key inflammatory regulator NLRP3, which is associated with oral and systemic diseases [107]. Inflammasomes are part of inflammatory complexes that respond to pathogens or cell damage by maturing proinflammatory cytokines [108]. Pathogens such as *T. forsythia*, *T. denticola*, *P. intermedia*, *Prevotella nigrescens*, *Prevotella micros*, and *F. nucleatum* are capable of long-term stimulation of the expression of toll-like receptors (TLRs), which recognize specific conserved molecular patterns associated with pathogens (PAMPs) [109,110]. For example, *P. gingivalis* endotoxin binds to CD14 molecules on the surface of macrophages and activates the TLR2/4 signaling pathway [111]. Stimulation of TLRs induces intracellular signaling cascades mediated by nuclear factor kappa B (NF-κB), which, when translocated to the nucleus, activates the transcription of genes encoding proinflammatory cytokines and chemokines, including IL-1β and TNFα [112]. Thus, a chronic systemic inflammatory response is accompanied by high levels of proinflammatory factors and tissue infiltration by immune cells, which can lead to progressive tissue damage and hemodynamic changes.

### 4.2. Hypertension

Many studies have demonstrated a correlation between various oral microorganisms and hypertension (HTN). For instance, an increased prevalence of the phylum *Firmicutes* has been observed in individuals diagnosed with hypertension. [113]. Additionally, there are established associations between hypertension and specific bacteria, including *Actinobacillus*, *Aggregatibacter*, *Atopobium*, *Bulleidia*, *Cupriavidus*, *Desulfomicrobium*, *Eikenella*, *Euzebya*, *Kingella*, *Moraxella*, *Olsenella*, *Pasteurella*, *Pelomonas*, and *Selenomonas*. Pathogens linked to periodontitis, such as *Porphyromonas*, *Fusobacterium*, and *Treponema*, are more abundant in the subgingival plaques of HTN participants and are positively correlated with IL-6 and/or CRP levels [114]. A recent study found that the mean systolic and diastolic blood pressure was positively associated with the bacterial load levels of *P. gingivalis*, *T. forsythia*, *A. actinomycetemcomitans*, and *T. denticola* [115]. The primary mechanism by which oral microbiome dysbiosis negatively impacts blood pressure is believed to be related to nitrate metabolism. Human nitrate reduction requires nitrate-reducing bacteria, since mammalian cells cannot effectively perform this function [116]. Oral commensal bacteria provide a significant metabolic role by serving as a nitric oxide synthase (NOS)-independent source of nitric oxide (NO). NO produced in the vasculature diffuses into the underlying smooth muscle, causing relaxation, vasodilation, decreased systemic blood pressure, and increased blood flow and oxygen delivery to certain vascular beds. In healthy individuals, endothelial nitric oxide synthase (eNOS) activation causes vasodilation in both muscle conduits and resistance arterioles. Conversely, in individuals with atherosclerosis or endothelial dysfunction, such stimulation results in decreased peripheral vasodilation and paradoxical vasoconstriction in the coronary arteries, indicating reduced NO production and bioavailability [117]. Recent evidence suggests a critical link between endothelium dysfunction and subsequent NO depletion in the development of hypertension and other cardiovascular disorders [118]. Plasma nitrite levels increase after nitrate administration, provided they can be reduced by oral nitrate-reducing bacteria. Salivary nitrate is metabolized to nitrite by a two-electron reduction during anaerobic respiration by nitrate reductases produced by facultative and obligate anaerobic commensal oral bacteria. This pathway, known as the enterosalivary nitrate–nitrite–nitric oxide pathway, may positively influence NO homeostasis and represents a potential symbiotic relationship between oral bacteria and their human hosts [119]. A metagenomic study identified several bacteria involved in nitrate metabolism, including *Granulicatella adiacens*, *Haemophilus parainfluenzae*, *Actinomyces odontolyticus*, *Actinomyces viscosus*, *Actinomyces oris*, *Neisseria flavescens*, *Neisseria mucosa*, *Neisseria sicca*, *Neisseria subflava*, *Prevotella melaninogenica*, *Prevotella salivae*, *Veillonella dispar*, *Veillonella parvula*, and *Veillonella atypica* [120]. However, another study showed that *Prevotella* and *Veillonella* were significantly more abundant in hypertensive groups and positively correlated with blood pressure and hyperglycemia [113]. Conversely, bacterial species typically associated with good oral health, such as *Neisseria subflava*, were more common in normotensive patients than in hypertensive patients [121]. This suggests that bacterial concentration could serve as a biomarker for hypertension development. A recent study revealed a higher prevalence of *Corynebacterium durum* in younger women (aged 50–59 years) compared to older women (aged ≥70 years). *C. durum* plays a role in the oxidation process and conversion of nitrite to NO. It is plausible that a decline in the number of this bacterium with age impairs the capacity to maintain normal blood pressure [115]. Furthermore, research has demonstrated that in individuals with ischemic stroke and hypertension, diminished oral nitrate-reducing bacterial levels are linked to more unfavorable stroke outcomes [122].

### 4.3. Endothelial Dysfunction

Endothelial dysfunction is known to play a crucial role in initiating and promoting vascular diseases such as hypertension, atherosclerosis, dyslipidemia, and type 2 diabetes [123]. The precise biological pathways through which periodontitis accelerates vascular disease development remain incompletely understood. However, three main hypotheses have been proposed: bacteriological, inflammatory, and immunological theories [124]. Periodontal microorganisms and their toxic products can enter the bloodstream from ulcerated areas during activities like chewing, brushing, or invasive dental treatments [125]. Research has shown that *P. gingivalis* invades and damages endothelial cells by affecting intercellular adhesion molecule 1 (ICAM-1) [126]. Additionally, *P. gingivalis* invasion of gingival epithelium and endothelial cells may be facilitated by *F. nucleatum* and *T. forsythia* [127]. *P. gingivalis* promotes cytokine production and stimulates Th1 cells, which increases macrophage activation and vascular inflammation. Studies have reported that gingipains and outer membrane vesicles of *P. gingivalis* increase vascular permeability by proteolytically cleaving platelet endothelial cell adhesion molecule 1 (PECAM-1) [124]. In an in vitro model of human coronary artery endothelial cells (HCAEC), treatment with *P. gingivalis* LPS resulted in significant production of MCP-1, IL-6, and granulocyte–macrophage colony-stimulating factor in HCAEC. In a similar model, the negative effect of *A. actinomycetemcomitans* LPS was more pronounced. The proposed mechanism is the activation of NF-κB and subsequent secretion of proinflammatory cytokines [128,129]. Periodontopathic bacterial HSP60 (GroEL) is homologous to host HSP and is highly immunogenic. Host T cells do not recognize the homology between host HSP60 expressed by endothelial cells (EC) and GroEL from periodontal microorganisms. Consequently, antibodies directed against bacterial GroEL cross-react with HSP60 on EC, leading to autoimmune reactions and endothelial dysfunction [130,131]. The recognition of noxious substances from the periodontium triggers the release of an inflammatory cytokine network, resulting in a complex proinflammatory and prothrombotic phenotype of endothelial cells. Patients with periodontitis have been reported to have higher plasma fibrinogen levels and white blood cell counts than controls [132]. Elevated fibrinogen can further stimulate the production of IL-6, IL-8, TNF-α, MCP-1, MMP-1, and MMP-9, worsening endothelial inflammation [133,134]. A recent study demonstrated that outer membrane vesicles (OMVs) of *P. gingivalis* can induce endothelial dysfunction via the Cyclic GMP-AMP synthase-stimulator of interferon genes–tank-binding kinase 1 (cGAS-STING-TBK1) signaling cascade [135]. Moreover, an in vitro study demonstrated that *P. gingivalis* has the potential to impair mitochondrial function in endothelial cells by regulating the activity of the RhoA/ROCK1 pathway [136].

The coagulation and fibrinolytic system, which includes fibrinogen, von Willebrand factor, tissue plasminogen activator (tPA), PAI-1, and coagulation factors VII and VIII, plays a vital role in maintaining vascular homeostasis [137]. PAI-1, a key fibrinolytic compound and a risk factor for vascular diseases, is significantly reduced by *P. gingivalis* in human endothelial cells. The degradation of lysine-specific gingipain-K (Kgp) leads to permeabilization and dysfunction of vascular endothelial cells through low-density lipoprotein receptor-related protein (LRP1). In vitro studies have also shown that LPS induces caspase-mediated cleavage of adherens junction proteins [138]. Most studies suggest that intensive periodontitis treatment, including oral hygiene education, scaling, and root planing, improves endothelial function. Potential biomarkers linking periodontitis to endothelial dysfunction, such as CRP, ICAM-1, IL-1, E-selectin, vWF, and PAT-1, have been found to decrease after periodontal treatment [139,140]. It is noteworthy that treatment with potassium nitrate effectively reduced endothelial dysfunction and proinflammatory cytokine levels in a mouse model of periodontitis [141].

### 4.4. Atherosclerosis

Atherosclerosis is currently recognized as a multifactorial disease caused not only by lipid accumulation but also by a subacute inflammatory state [142]. Epidemiological data confirm that individuals with periodontitis have a higher prevalence of subclinical cardiovascular disease, peripheral arterial disease, and coronary events [143]. Furthermore, data exist regarding the impact of periodontitis on the arterial stiffness index, with the influence dependent on the severity of the condition [144]. A study involving 831 individuals demonstrated that patients with apical periodontitis had elevated levels of proinflammatory cytokines (hsCRP, IL-6) and E-selectin. Several signaling pathways associated with the inflammatory response are implicated in atherosclerosis, including the NLRP3 inflammasome, Toll-like receptors, proprotein convertase subtilisin/kexin type 9, and Notch and Wnt signaling pathways [145]. These pathways play a crucial role in the development and progression of atherosclerosis [146,147]. Additionally, other molecules that promote the inflammatory response, such as ICAM-1, VCAM-1, and P-selectin, are also involved. The L-, P-, and E-selectin adhesion molecule families contain an N-terminal lectin-like domain followed by an epidermal growth factor (EGF)-like domain and varying numbers of repeating units homologous to complement EGF/protein regulators. An unproven theory, the “Trojan horse theory”, suggests that pathogens can spread through phagocytosis mediated by circulating immune cells and evade microbial killing [148]. These pathogens can be found in affected vascular tissue. DNA, RNA, or antigens from *P. gingivalis*, *A. actinomycetemcomitans*, or *Veillonella* species have been detected in atheromatous samples, suggesting that multiple pathogenic species may invade the affected area [149]. A study showed that *P. gingivalis* accelerates the progression of atherosclerosis in ApoE−/− mice by activating the TLRs-NF-κB signaling axis, suppressing basic helix–loop–helix ARNT-like protein 1 (BMAL1) transcription, releasing the circadian protein CLOCK and thereby increasing oxidative stress and inflammatory response in aortic endothelial cells [150]. Furthermore, the influence of *P. gingivalis* on the expression of glycogen synthase kinase 3 beta (GSK-3β)/nuclear factor (erythroid-derived 2)-like 2 (Nrf2)/tetrahydrobiopterin (BH4)/NOS may result in the impairment of the antioxidant response and vascular relaxation capacity [151,152]. There is evidence that *P. gingivalis* induces the oxidation of LDL and high-density lipoprotein (HDL), promoting lipid accumulation in the vascular wall. Furthermore, gingipains are capable of inducing lipid peroxidation [153]. Macrophages are the primary inflammatory cells in atherosclerotic lesions, exhibiting both proinflammatory (M1) and anti-inflammatory (M2) phenotypes. It has been shown that periodontal infection can enhance the phenotypic transition of M1/M2 macrophages to M1 macrophages [154]. Additionally, a clinical study found that periodontitis therapy significantly reduced levels of inflammatory factors IL-6, IL-8, and CRP in patients with atherosclerotic cardiovascular disease with high CRP levels (CRP ≥ 3 mg/L). However, no significant differences were observed in serum levels of IL-10, IL-1β, IFN-γ, and TNF-α before and after treatment [155]. These findings highlight the significant impact of the oral microbiome on the development of vascular damage and atherosclerosis (Figure 2).

### 4.5. Metabolic Syndrome, Obesity, and Metabolic Disorders

Metabolic syndrome is a disorder characterized by a combination of factors such as hypertension, dyslipidemia, high plasma glucose, and central obesity. Three of these five components must be present together to make a diagnosis of the syndrome [156]. Metabolic syndrome is a complex cluster of interrelated conditions that occur simultaneously and significantly elevate the risk of developing cardiovascular disease and type 2 diabetes [157]. The presence of periodontitis has been shown to increase the risk of developing metabolic syndrome. At the same time, individuals with metabolic disorders are 38% more likely to develop periodontitis [158]. A study of 1023 individuals revealed a correlation between the depth of periodontal pockets and one or more metabolic components over a four-year period. Furthermore, the results demonstrated that the abundance of *Spirochaete*, *Tenericutes*, and *Coriobacteriales* was higher in patients with metabolic syndrome, whereas levels of *Betaproteobacteria* and *Corynebacteriales* were higher in healthy individuals [159]. Moreover, a study of 228 individuals from the Korean Epidemiological Genome Study revealed that the oral microbiome of participants with metabolic syndrome exhibited a higher prevalence of *Granulicatella* and *Neisseria* and a lower prevalence of *Peptococcus* [160]. Another study demonstrated a correlation between metabolic syndrome and the extent of alveolar bone loss [161]. Moreover, a substantial body of research has demonstrated a correlation between the oral microbiome and a range of pathological conditions, including periodontitis, as well as specific components of the metabolic syndrome, such as obesity, dyslipidemia, and carbohydrate metabolism disorders [162,163,164]. Numerous studies have shown that type 1 diabetes plays a role in the development and severity of periodontitis [165]. This association is largely due to both levels of proinflammatory cytokines and genetic factors [166]. A metagenomic investigation of the oral microbiome of patients with type 2 diabetes revealed that obese participants exhibited reduced microbial diversity and, an increased prevalence of *Proteobacteria*, *Chloroflexi*, and *Firmicutes*, with a notable absence of members belonging to the *Bacteroidetes* phylum. Furthermore, research on patients with periodontitis and diabetes mellitus revealed disparities linked to obesity [167]. Another study showed that the prevalence of *Bifidobacterium* and *S. mutans* was higher in overweight or obese subjects compared to those with a normal body mass index. Furthermore, a positive correlation was identified between body fat accumulation and the quantity of *Bifidobacterium* present in saliva, suggesting a potential interaction between oral microbial communities and weight gain [168]. Moreover, the study demonstrated that the prevalence of *A. actinomycetemcomitans* was significantly higher in individuals with diabetes compared to those without the condition. Furthermore, no significant differences were observed in the prevalence of *P. gingivalis*, *T. forsythia*, *P. intermedia*, and *F. nucleatum* between the groups [169]. It has been demonstrated that elevated levels of proinflammatory cytokines and mediators in periodontitis can influence lipid metabolism. This, in turn, may result in alterations to lipid and lipoprotein homeostasis, a process that has been linked to inflammatory conditions in other systems. The synthesis of LDL, adipose tissue lysis, increased de novo fatty acid synthesis in the liver, suppression of fatty acid oxidation, and disruption of LDL clearance are among the processes that contribute to the development of inflammatory dyslipidemia [159]. A recent study demonstrated that an intravenous injection of sonicated *P. gingivalis* resulted in altered endocrine function of brown adipose tissue in mice. The authors concluded that endotoxemia by *P. gingivalis* has the potential to affect obesity by disrupting brown adipose tissue function [170]. The precise mechanisms of bidirectional communication between oral microorganisms and metabolic disorders remain unclear. Adipose tissue is capable of responding to inflammatory processes and producing proinflammatory cytokines, including visfatin, leptin, and resistin, as well as anti-inflammatory mediators, such as adiponectin. It has been observed that, in obese individuals, the synthesis of proinflammatory adipokines in adipose tissue increases while the synthesis of anti-inflammatory adipokines, such as adiponectin, decreases [171]. Furthermore, the deterioration of the equilibrium between reactive oxygen species and antioxidants in both obesity and periodontitis results in augmented oxidative stress, which in turn contributes to a shift in the immune response toward hyperinflammation. Moreover, the significance of maintaining equilibrium in the immune response is exemplified by the observation that lipoxin (Lipoxin A4) levels, which are instrumental in resolving inflammation, are diminished in individuals with metabolic syndrome. Additionally, there is a negative correlation between lipoxin levels and both periodontal parameters and metabolic syndrome [172]. Furthermore, there is evidence suggesting that the oral microbiome plays a role in the development of liver diseases, particularly NAFLD. A recent investigation demonstrated that the oral administration of *P. gingivalis* or *P. intermedia* in a mouse model of NAFLD fed a choline-deficient, high-fat diet resulted in alterations to the composition of the gut microbiota and serum metabolome. These alterations subsequently modified the liver transcriptome toward a more pathogenic form of NAFLD [173].

## 5. Oral–Brain and Oral–Gut–Brain Axis

Two primary routes for oral bacteria and their metabolites to penetrate the brain have been identified in the scientific literature. Firstly, microorganisms can access the systemic bloodstream via damaged vessels within periodontal pockets [174]. This infection contributes to the dysfunction of the blood–brain barrier endothelium, likely through the degradation of intercellular contacts, particularly tight junction protein 1 (ZO-1) [175]. Experimental animal studies have demonstrated the possibility of periodontal pathogens penetrating the brain through the hematogenous route [176]. Secondly, periodontal microorganisms or their metabolic products can access the central nervous system via peripheral nerves, including the glossopharyngeal and trigeminal nerve [177]. There is a bidirectional relationship between chronic periodontitis and neurodegenerative diseases. Studies have shown that periodontitis is associated with reduced cognitive status, primarily due to chronic inflammation [178,179]. Periodontal pathogens such as *P. gingivalis* and *F. nucleatum* are linked to central nervous system disorders and neuropsychiatric diseases [180,181]. An experimental periodontitis model caused by *P. gingivalis* showed a negative impact on Alzheimer’s disease progression, attributed to increased beta-amyloid production and elevated levels of proinflammatory cytokines [182,183]. Treatment with low-molecular gingipain inhibitors reduced these negative impacts [184]. Patients with mild cognitive impairment (MCI) and periodontitis exhibited a more pronounced decline in memory function compared to those without periodontitis [185]. Additionally, *P. gingivalis* has been observed to promote inflammatory destruction of the synovium through protein citrullination, leading to the production of antibodies against citrullinated peptides and the activation of autoreactive T cells [186].

Traditionally, the microbial colonization of the oral cavity is thought to occur after birth. However, recent research shows that the microbiome begins to form before birth [187]. The amniotic fluid of 70% of pregnant women has been found to contain microorganisms, including several species of oral microbes such as *Streptococcus*, *Fusobacterium*, *Neisseria*, *Prevotella*, and *Porphyromonas* [188,189]. A population-based cohort study was conducted to characterize the placental microbiome, utilizing placental samples from 320 subjects. It is noteworthy that the placental microbiome exhibits striking similarities to the oral microbiome yet displays minimal overlap with other niches, such as the genitourinary tract or intestine [190]. The likelihood of children of parents with periodontitis being colonized by bacteria that cause periodontitis, including *P. gingivalis*, *A. actinomycetemcomitans*, *S. parasanguinis*, and *F. nucleatum*, is greater than that of children of parents without periodontitis. This trend persists despite plaque control, indicating that individual oral hygiene practices are inadequate for preventing microbial inheritance [191]. Few studies have also demonstrated the effects of periodontal pathogens on brain development in vivo. Following the administration of *P. gingivalis* to pregnant rats, the bacteria were observed to be present throughout the brain, with a particular concentration in the hippocampus. Additionally, elevated levels of calcium-binding adapter protein 1-positive microglia (IBA1^+^) and an increased number of glial fibrillary acidic protein-positive astrocytes (GFAP^+^), which are markers of neuroinflammation, were observed [192]. Another study demonstrated that intragingival administration of *P. gingivalis* LPS to female rats resulted in impaired spatial learning and memory in the offspring when compared to the control group [193].

The initial colonization of the intestines occurs via the oral cavity. However, even in adulthood, oral bacteria can enter the intestines with saliva. The evidence regarding the potential for colonization of the gut microbiome by oral microorganisms in healthy individuals is inconclusive [194,195,196]. Nevertheless, studies have demonstrated that the integrity of the intestinal barrier is compromised with advancing age and the presence of diseases such as alcoholism, liver cirrhosis, inflammatory bowel disease, colorectal cancer, and rheumatoid arthritis [196,197,198,199,200,201]. Furthermore, the colonization of the intestine by oral bacteria has been observed in patients with gastric achlorhydria caused by long-term use of proton pump inhibitors (PPIs). Long-term PPI therapy in patients with gastroesophageal reflux disease (GERD) has been observed to result in a higher abundance of oral bacteria in the gut than in healthy individuals [202,203]. The primary impediment to the incursion of oral microorganisms into the intestine is the resistance to colonization provided by the harmonious microbial configuration typical of a healthy intestine [204].

The presence of pathological conditions in the oral cavity, particularly periodontitis, has been observed to result in a notable increase in the prevalence of pathobionts within the saliva [205,206]. It has been demonstrated that experimental periodontitis can result in intestinal dysbiosis, inflammation, and increased intestinal barrier permeability [207]. Thus, the introduction of *P. gingivalis* via gavage in mice resulted in an inflammatory condition of the intestine, characterized by alterations in the composition of the intestinal microbiota and disruption of the intestinal barrier function [208]. Another study demonstrated that periodontitis may influence the *Firmicutes/Bacteroidetes* ratio of the gut microbiome [209]. Furthermore, in a rodent model of periodontitis caused by *F. nucleatum*, researchers observed the migration of oral bacteria into the intestinal tract, accompanied by alterations in the gut microbiota [210]. In particular, changes occurred in the gut ecosystem, with a notable decrease in *Verrucomicrobia* and an increase in *Proteobacteria* evident over the two-week period, followed by an increase in *Bacteroidetes* and *Firmicutes* observed at four weeks [211]. A clinical study demonstrated changes in the gut microbiota in patients with periodontitis, exhibiting an increase in the abundance of the phyla *Proteobacteria*, *Firmicutes*, *Euryarchaeota*, and *Verrucomicrobia* in comparison to healthy individuals [209]. Another study revealed a correlation between *Veillonella* concentrations in saliva and feces, suggesting intestinal colonization by these bacteria [114]. Remarkably, oral microorganisms can alter the gut microbiome by affecting food intake. For instance, high levels of *Streptococcus* and *Lactobacillus* in saliva are thought to be associated with a reduction in the perception of sourness because of the production of organic acids by these bacteria, which elevate the taste threshold [212]. It has also been shown that T. forsythia may modify the taste buds of the tongue, predisposing to the consumption of fatty foods [213].

The intestines and the central nervous system (CNS) maintain bidirectional communication with each other. The vagus nerve, a major component of the parasympathetic nervous system, plays a role in the transmission of information between the gut and the central nervous system [214]. Consequently, the vagus nerve fulfills a number of functions. Firstly, it transmits a signal from the brain to the intestines via efferent nerve fibers. Secondly, it detects metabolites and hormones derived from the microbiota, which are released by endocrine cells of the intestine via afferent nerve fibers [215]. Enterochromaffin cells of the intestine have been observed to possess Toll-like receptors, enabling them to respond to intestinal dysbiosis. Metabolites derived from the gut microbiota (γ-aminobutyric acid, norepinephrine, dopamine, serotonin, tyramine, and tryptophan) can act on brain cells directly or indirectly by acting on vagal afferent fibers [216]. It has been demonstrated that metabolites produced by the intestinal microbiota are capable of penetrating the intestinal mucosal layer and entering the bloodstream, subsequently crossing the blood–brain barrier and regulating microglial function [217]. Peripheral cytokines transmit inflammatory signals to the brain via afferent nerves. The study demonstrated that germ-free (GF) mice exhibited pervasive deficiencies in microglial maturation and function, leading to the suppression of innate immune responses. It is established that certain bacterial strains are capable of disrupting the biosynthesis and metabolism of neurotransmitters [218].

The evidence regarding the impact of gut microbial diversity on stroke remains inconclusive and controversial [219,220]. A number of studies have shown a higher prevalence of the phyla *Actinobacteria*, *Proteobacteria*; class *Gammaproteobacteria*; families *Bacteroidaceae*, *Bifidobacteriaceae*, *Enterobacteriaceae*, *Lachnospiraceae*, *Porphyromonadaceae*, *Prevotellaceae*, *Rikenellaceae*, *Ruminococcaceae*, and *Veillonellaceae*; the genera *Bacteroides*, *Escherichia/Shigella*, *Lactobacillus*, *Prevotella*, *Ruminococcus*, and *Streptococcus*, and decreased levels of *Bacteroidetes* and *Firmicutes*; and the genera *Eubacterium*, *Faecalibacterium*, and *Roseburia* in stroke patients compared to healthy controls [221]. Additionally, increased abundance of *Proteobacteria* and decreased abundance of *Bacteroides* were associated with stroke severity [222]. A recent study demonstrated that an increased abundance of *Bacteroides pectinophilus* is associated with a 28% lower risk of cardioembolic stroke in both European and Asian populations [223]. Another study found that the abundance of *Proteobacteria* and *Fusobacteria* was significantly higher in participants with ischemic and hemorrhagic stroke, respectively. The abundance of *Butyricimonas*, *Alloprevotella*, and *Escherichia* was significantly higher in people with ischemic stroke than in healthy individuals [224]. There is evidence of increased abundance of *Akkermansia* as well as *Lactobacillaceae*, *Enterobacteriacea*, and *Porphyromonadaceae* in cases of acute ischemic stroke, especially in severe stroke [225]. A deficiency in short-chain fatty acid producers following an ischemic stroke has been observed to be associated with an increase in interleukin (IL)-17+ γδ T cells and a decrease in regulatory T cells in animal models [226]. Concurrently, a clinical study revealed markedly elevated levels of SCFA-producing genera *Odoribacter* and *Akkermansia* in patients diagnosed with ischemic stroke. Additionally, research demonstrated that a diet rich in dietary fiber was linked to a reduced risk of stroke [227,228]. A vegetarian diet was also found to be associated with a lower risk of stroke compared to a diet high in protein. Nevertheless, the currently available evidence is insufficient to reach a definitive conclusion [229].

Dysbiosis of the oral microbiome exerts a complex influence on the host’s body. Microorganisms that penetrate the systemic bloodstream and cranial nerves into the central nervous system contribute to neuroinflammation, accumulation of pathological proteins, and disruption of the BBB. Pathogenic microorganisms that enter the intestines with saliva can cause dysbiosis and changes in microbial diversity. The repression of certain microbial growth and the promotion of others result in alterations in the production of SCFAs and neurotransmitters.

## 6. Current Limitations and Future Direction

The current evidence base for the link between oral health and stroke is mainly derived from studies of oral diseases such as periodontitis and dental caries. This is due to the high number of such studies and the large sample sizes, which have included hundreds of thousands and even millions of participants [47,72]. Concurrently, studies investigating the relationship between the oral microbiome and stroke at the level of diversity or individual taxa remain scarce, comprising only a few studies with tens or hundreds of participants [86,87]. This circumstance makes it challenging to reach definitive conclusions about the causal relationship and the broader implications of the results. A further understanding of the oral microbiome as a risk factor for stroke in young adults without significant periodontal disease could also prove beneficial [230]. Additionally, much of the research has been conducted in specific geographic regions or among certain demographic groups, which limits the generalizability of the results. Differences in diet, lifestyle, socioeconomic status, availability of health care, medication use and genetic factors across populations can influence the composition of the oral microbiome and its relationship with stroke, underscoring the need for more diverse and representative studies [231].

While associations between oral microbiota and stroke have been observed, the underlying mechanisms remain poorly understood. It is unclear how specific bacterial species or microbial communities contribute to stroke pathogenesis. Further research into the oral-brain and oral–gut–brain axes represent a promising area of study. The transfer of oral bacteria to the intestine has the potential to alter the intestinal microbiome, which in turn may affect overall health [232]. The negative impact of certain oral bacteria on neurodegenerative diseases, such as Alzheimer’s and Parkinson’s, potentially may share common mechanisms with stroke severity and influence rehabilitation outcomes [233]. The relationship between microorganisms, both in planktonic form and in the composition of biofilms, is a crucial area of study. The interaction between different species has the potential to activate the growth of pathogens, enhance virulence, and contribute to the strengthening of the systemic negative effect of oral dysbiosis [234].

More research is needed to elucidate the pathways through which oral bacteria influence vascular health, inflammation, and other stroke-related risk factors. Integrating multi-omics approaches, including genomics, transcriptomics, proteomics, and metabolomics, can provide a more comprehensive understanding of the interactions between the oral microbiome and stroke [235,236]. Combining these data with clinical and environmental factors will help identify key biomarkers and therapeutic targets.

## 7. Conclusions

The complex interconnection between the dysbiosis of the oral microbiome and stroke underscores a crucial yet frequently disregarded facet of cardiovascular health. This article synthesizes current knowledge on how alterations in the oral microbiome can influence stroke development through various mechanisms. Evidence indicates that microorganism-associated oral diseases, such as periodontitis, play a crucial role in exacerbating stroke risk factors, including chronic inflammation, dyslipidemia, hypertension, obesity, atherosclerosis, and endothelial dysfunction. Several studies have linked the species composition and diversity of the oral microbiome to stroke risk. However, more studies are needed to show the relationship between modifying the oral microbiome and reducing stroke risk and its effect on stroke risk factors.

A deeper comprehension of the intricate interrelationships between the oral microbiome and overall health status paves the way for the development of innovative preventative and therapeutic approaches. Future research should aim to delineate the precise pathways through which oral bacteria contribute to stroke pathogenesis and explore targeted interventions to restore microbial balance. Integrating oral health into comprehensive stroke prevention programs could significantly reduce the burden of this debilitating disease. By bridging the gap between dental and medical health practices, we can pave the way for more effective, holistic approaches to stroke prevention and overall well-being.

## Figures and Tables

**Figure 1 microorganisms-12-01732-f001:**
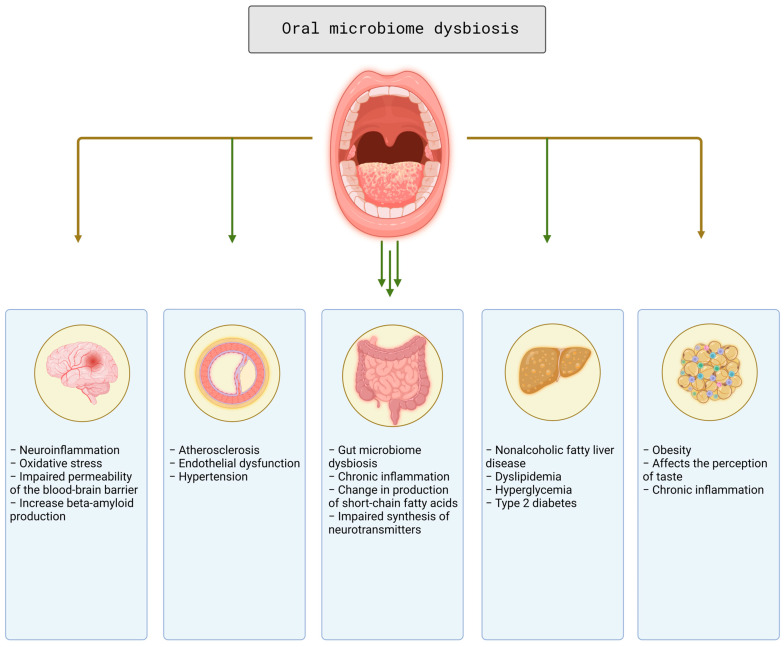
Schematic representation of how oral microbiome can increase the risk of stroke. Major negative effects of dysbiosis on the central nervous system, vascular, gut, liver, lipid, and carbohydrate metabolism, and adipose tissue are presented. Created with BioRender.com (accessed on 15 July 2024).

**Figure 2 microorganisms-12-01732-f002:**
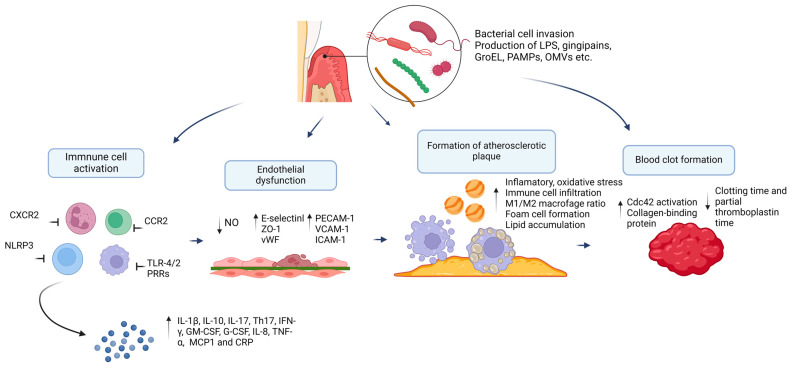
Schematic representation of the mechanisms of influence of oral microbiome dysbiosis on inflammation, endothelial dysfunction, atherosclerotic plaque formation, and thrombus formation. Microorganisms invade the vascular wall and tissue of atherosclerotic plaque and thrombus both directly and via the secretion of virulence factors and hyperactivation of the immune system. Created with BioRender.com (accessed on 15 July 2024).

**Table 2 microorganisms-12-01732-t002:** Recent studies investigating the association of oral microbiome with stroke.

Bacteria	Stroke or Associate Condition	Year	Population	Brief Results	Reference
*Porphyromonas gingivalis* and *Aggregatibacter actinomycetemcomitans*	Stroke	2012	12 patients with ischemic or hemorrhagic stroke and 60 systemically healthy individuals	Participants with stroke had higher rates of pocket depth, gingival bleeding, plaque index, and number of missing teeth. In addition, the number of *P. gingivalis* was higher in the stroke group.	Ghizoni et al. [97]
*Porphyromonas gingivalis*	Stroke	2024	63 individuals with stroke included in the multicentre compoCLOT cohort study	*P. gingivalis* was found in 33.7% of thrombi in stroke patients. *P. gingivalis* was associated with lower rates of complete reperfusion, favorable stroke outcomes, and higher levels of neutrophil elastase in thrombi.	Seckendorff et al. [91]
16 periodontal pathogens	Stroke	2020	534 patients who had experienced an acute stroke	The antibody level of *F. nucleatum* was identified as an independent predictor of unfavorable stroke outcomes.	Nishi et al. [92]
9 periodontal pathogens	Stroke	2020	639 patients who had experienced an acute stroke	A positive titer of serum antibodies against *Campylobacter rectus* has been associated with the presence of cerebral microbleeds in patients with acute stroke.	Shiga et al. [93]
*A. actinomycetemcomitans*, *P. gingivalis*, and *P. intermedia*	Cerebral infarction	2013	8 individuals with diabetic nephropathy and 13 with non-diabetic nephropathy	A higher prevalence of *A. actinomycetemcomitans* was observed in patients with diabetic nephropathy compared to those with non-diabetic nephropathy. A significantly higher incidence of cerebral infarction was observed in patients with diabetic nephropathy compared to patients with non-diabetic nephropathy.	Murakami et al. [98]
*Streptococcus mutans*	Cerebral microbleeds	2020	111 individuals from 63 to 81 years old	CNM-expressing *S. mutans* was associated with a higher risk of cerebral microbleeds.	Hosoki [83]
Metagenome	Ischemic stroke	2024	81 participants who had undergone a thrombectomy procedure	The investigation revealed that the abundance of *Bacillus*, *Parabacteroides*, *Prevotella*, *Streptococcus*, *Romboutsia*, *Corynebacterium*, and *Roseburia* was higher in thrombus tissue than in arterial blood. Furthermore, the results indicated that alcohol consumption was associated with a higher bacterial load.	Wang et al. [99]
*Porphyromonas gingivalis*	Thrombosis	2019	16 systemically healthy, 8 patients with healthy periodontium, and 8 patients with periodontitis	*P. gingivalis* LPS-treated platelets showed significantly greater spreading and a greater number of filopodia than controls. LPS stimulation of platelets promoted Cdc42 activation. Exposure to LPS significantly reduced both the clotting time and partial thromboplastin time.	Senini et al. [90]
*Streptococcus mitis*, *Porphyromonas gingivalis*, and *Aggregatibacter actinomycetemcomitans*	Ischemic stroke	2019	75 patients with acute ischemic stroke	The relative amount of *Streptococcus* species DNA was 5.1 times higher in thrombi compared to blood samples. All thrombi were negative for both *P. gingivalis* and *A. actinomycetemcomitans*.	Patrakka et al. [84]
*Streptococcus sanguinis*, *Streptococcus mitis*, and *Streptococcus gordonii*	Ischemic stroke	2023	61 patients with ischemic stroke	84.8% of thrombi, 80.0% of carotid endarterectomy specimens, and 31.3% of carotid artery specimens were positive for *Streptococcus*. Most *Streptococcus* were found within neutrophils, but some samples also had remnants of bacterial biofilm as well as free bacterial infiltrates.	Patrakka et al. [86]
*Streptococcus*, *P. gingivalis*, and *A. actinomycetemcomitans*	Ischemic stroke	2021	71 patients with acute ischemic stroke	Oral streptococcal DNA was detected in 78.9% of thrombus aspirates. Patients with the best oral health had more oral streptococcal DNA in the thrombus than those in the worst pathology group. There was a trend for ≥50% carotid artery stenosis to be associated with more severe dental pathology.	Patrakka et al. [85]
*Pseudomonas aeruginosa*, *Klebsiella pneumoniae*, and *Escherichia coli*	Stroke	2020	102 patients with acute stroke	An increase in the bacteria *P.aeruginosa*, *K. pneumoniae*, and *E. coli* was observed one month after the stroke.	Perry et al. [78]
Oral metagenome	Stroke	2023	262 patients with stroke	Lower levels of *Streptococcus* have been demonstrated to be associated with stroke-associated pneumonia. The enrichment of specific taxa within the phylum *Actinobacteriota* was identified as an independent risk factor for poor 30-day clinical outcomes.	Ren et al. [87]
Oral metagenome	Cryptogenic ischemic stroke	2024	308 individuals from young patients diagnosed with cryptogenic ischemic stroke	The predominant microbiota in saliva is largely similar between cases and controls. At the species level, the abundance of *B. goodwinii*, *V. boronicumulans*, *Pseudomonas* sp. *AN-B15*, *Actinoalloteichus* sp. *GBA129-24*, and *T. arsenitoxydans* were higher in the stroke group. The levels of *M. alkalescens*, *Streptomyces* sp. *LBUM 1475*, *P. dioxanivorans*, *A. dokdonella*, and *F. frigiditurris* were lower in the stroke group than in the control group.	Manzoor et al. [79]
Oral metagenome	Ischemic stroke	2023	146 subjects	The stroke and high-risk stroke groups had higher alpha diversity indices for Chao1, Shannon, Simplons, and the genera *Streptococcus*, *Prevotella*, *Veillonella*, *Fusobacterium*, and *Treponema*.	Sun et al. [74]
*Porphyromonas gingivalis*, *Tanarella forsythia*, and *Treponema denticola*	Coronary artery disease	2021	80 patients scheduled for coronary artery bypass grafting or angioplasty	In 10%, 12.5%, and 1.3% of the atherosclerotic plaque samples, *P. gingivalis*, *T. forsythia*, and *T. denticola* were detected.	Rao et al. [94]
*Aggregatibacter actinomycetemcomitans*, *Tannerella forsythia*, *Porphyromonas gingivalis*, and *Treponema denticola*	Coronary artery disease	2013	51 patients with chronic periodontitis	In 0%, 31.4%, 45.1%, 39.2%, and 51% of the atherosclerotic plaque samples, *A. actinomycetemcomitans*, *T. forsythia*, *P. gingivalis*, and *T. denticola* were detected.	Mahendra et al. [95]
Oral metagenome	Ischemic stroke	2023	143 patients with ischemic stroke and 143 healthy individuals	Alpha diversity was inversely associated with the risk of ischemic stroke. Furthermore, *Corynebacterium*, *Lautropia*, and *Selenomonas* were associated with an increased risk of stroke.	Wang et al. [75]
Oral metagenome	Multimorbidity including stroke	2023	201 adult subjects, including 84 with severe periodontal	The oral levels of the bacteria *Porphyromonas gingivalis* and *Tannerella forsythia* were demonstrated to be reduced in participants with multimorbidity, including stroke.	Shen et al. [96]

## Data Availability

Not applicable.

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
