# Peer review of "Oral Microbiome Dysbiosis as a Risk Factor for Stroke: A Comprehensive Review"

_microorganisms, 2024, doi:10.3390/microorganisms12081732_

Round 1
Reviewer 1 Report
Comments and Suggestions for Authors
Dear Authors,
I carefully read your paper and I congrats for the good work you've done.
Introduction, although generally satisfaying, could be improved adding some more info about IL role in inflammation and systemic diseases. I left a couple of paper that you can take as example and eventually include in your work
https://pubmed.ncbi.nlm.nih.gov/36141674/
https://pubmed.ncbi.nlm.nih.gov/38397350/
https://pubmed.ncbi.nlm.nih.gov/36621868/
I feel that conclusions could emphasize the need of systematic reviews/further studies to better understand the role between oral dysbiosis and systemic diseases
Author Response
The authors very much appreciated the constructive comments on this manuscript by the reviewer. The comments have been very thorough and useful in improving the manuscript.
Comment 1: Introduction, although generally satisfaying, could be improved adding some more info about IL role in inflammation and systemic diseases. I left a couple of paper that you can take as example and eventually include in your work.
Response 1: Yes, you are right. This information would be useful in this context. We have made the appropriate changes to the manuscript.
Comment 2: I feel that conclusions could emphasize the need of systematic reviews/further studies to better understand the role between oral dysbiosis and systemic diseases
Response 2: Corrected. We have made the appropriate changes to the manuscript.
We also re-structured the manuscript and made a number of additional changes.
Reviewer 2 Report
Comments and Suggestions for Authors
The article “Oral Microbiome Dysbiosis as a Risk Factor for Stroke: A Com-2 prehensive Review” deals with an incitant subject. However the following drawbacks have been identified:
Treatment options for stroke from introduction are not necessarily.
-row 70-71 “The most prevalent oral diseases, including dental caries, gingivitis, periodontitis, and oral cancer, are directly or indirectly associated with the activities of microorganisms [20,21].” -should be reformulated as microorganisms are primary etiology of periodontitis and caries.
-row 72 – the term “periodontal disease” includes many periodontal pathologies and so it should be replaced with “periodontitis” all over the manuscript. The two terms should not be used interchangeably.
-the sentence beginning at row 78 – oral diseases also influence the microbiome composition.
-fig 1 – the abbreviations are not explained in the figure’s legend. Please mention the provenience of the drawings, eventual the permission of use.
-row 112-115 -the sentence is not interesting here. Eventually, it could be modified and used in the introduction to highlight the systemic influences of periodontitis.
-row 116-143 – the paragraph should be removed since the subject of this review is not pregnancy-periodontitis links!!!!
-row144-165 –excessive information on pregnancy-periodontitis should be removed, maintained only data on pregnancy modifications as stroke risk factors and the information on the influence of periodontopathogens on brain development resumed.
The bacteria abbreviated name should be provided after providing its entire denomination and only after that used abbreviated. After the abbreviated use of the name of the bacterium, its full name will no longer appear in the manuscript
Practically the subject of section 2 was not treated.
-row 205-208 – erase the phrase which does not fit in this context!
-row 220- we don’t’ know what is it “dental caries episodes (≥4)”
The information in section 3 must be presented in a more coherent and concise manner.
Please explain the term” alpha diversity”
-row 303 - The information about periodontopathogenic bacteria in periodontitis has already been presented in section 3. It needs to be consolidated in one place and not repeated in each section in different ways.
Each table should be mentioned in the text.
Section 6 should better be entitled as microbiome- stroke risk factors link. The information in section 6 is redundant and repetitive with what was previously presented, so the entire manuscript needs to be reviewed, summarized, and appropriately divided into subchapters to avoid the repetitive presentation of information.
Comments on the Quality of English LanguageModerate revision is necessary.
Author Response
The authors very much appreciated the constructive comments on this manuscript by the reviewer. The comments have been very thorough and useful in improving the manuscript.
Comment 1:Treatment options for stroke from introduction are not necessarily.
Response 1: Corrected. We have made the appropriate changes to the manuscript.
Comment 2: -row 70-71 “The most prevalent oral diseases, including dental caries, gingivitis, periodontitis, and oral cancer, are directly or indirectly associated with the activities of microorganisms [20,21].” -should be reformulated as microorganisms are primary etiology of periodontitis and caries.
Response 2: Corrected. We have made the appropriate changes to the manuscript.
Comment 3:-row 72 – the term “periodontal disease” includes many periodontal pathologies and so it should be replaced with “periodontitis” all over the manuscript. The two terms should not be used interchangeably.
Response 3: Corrected. We have made the appropriate changes to the manuscript
Comment 4: -the sentence beginning at row 78 – oral diseases also influence the microbiome composition.
Response 4: Corrected. We have made the appropriate changes to the manuscript
Comment 5:-fig 1 – the abbreviations are not explained in the figure’s legend. Please mention the provenience of the drawings, eventual the permission of use.
Response 5: Corrected. We have made the appropriate changes to the manuscript. Information about the digital license is provided in unpublished materials.
Comment 6:-row 112-115 -the sentence is not interesting here. Eventually, it could be modified and used in the introduction to highlight the systemic influences of periodontitis.
Comment 7:-row 116-143 – the paragraph should be removed since the subject of this review is not pregnancy-periodontitis links!!!!
Comment 8:-row144-165 –excessive information on pregnancy-periodontitis should be removed, maintained only data on pregnancy modifications as stroke risk factors and the information on the influence of periodontopathogens on brain development resumed.
Response 6,7,8: We have removed this section. The part describing in vivo studies investigating the effects of oral bacteria on offspring brain health has been moved to Section 5.
Comment 9: The bacteria abbreviated name should be provided after providing its entire denomination and only after that used abbreviated. After the abbreviated use of the name of the bacterium, its full name will no longer appear in the manuscript
Response 9: Corrected. We have made the appropriate changes to the manuscript
Comment 10: Practically the subject of section 2 was not treated.
-row 205-208 – erase the phrase which does not fit in this context!
Response 10: Corrected. We have made the appropriate changes to the manuscript
Comment 11: -row 220- we don’t’ know what is it “dental caries episodes (≥4)”
Response 11: Corrected. We have made the appropriate changes to the manuscript
Comment 12: The information in section 3 must be presented in a more coherent and concise manner.
Please explain the term” alpha diversity”
Response 12: Corrected. We have made the appropriate changes to the manuscript
Comment 13: -row 303 - The information about periodontopathogenic bacteria in periodontitis has already been presented in section 3. It needs to be consolidated in one place and not repeated in each section in different ways.
Response 13: Corrected. We have made the appropriate changes to the manuscript.
Comment 14:Each table should be mentioned in the text.
Response 14: Corrected. We have made the appropriate changes to the manuscript.
Comment 15: Section 6 should better be entitled as microbiome- stroke risk factors link. The information in section 6 is redundant and repetitive with what was previously presented, so the entire manuscript needs to be reviewed, summarized, and appropriately divided into subchapters to avoid the repetitive presentation of information.
Response 15: Corrected. We have made the appropriate changes to the manuscript.
We also re-structured the manuscript and made a number of additional changes.
Reviewer 3 Report
Comments and Suggestions for Authors
It presents data on the association between microorganism-associated oral diseases and stroke development, as well as the mechanisms of bacterial influence on stroke risk factors, including chronic inflammation, dyslipidemia, hypertension, obesity, atherosclerosis, and endothelial dysfunction
The paper is original and interesting, has technical quality, is important in its field, and has an adequate style and overall representation. The structure should be improved for a more natural and easier flow of information. It is excessively long and should be more focused.
The title accurately represents the manuscript's contents, clearly reflecting the focus on the relationship between the oral microbiome and stroke.
The summary is clear but could be more concise by eliminating repetitions and simplifying the objective, which is the relationship between the oral microbiome and stroke
Identifies specific bacterial genera associated with oral disease periodontitis and caries Addresses the importance of dental health in stroke prevention. Yes, this is a new and original contribution, offering fresh insights into the connection between oral health and stroke risk factors. Correlation between periodontitis and stroke is not universal; some studies show no significant
Other confounding factors, such as socioeconomic status and access to healthcare, are not deeply explored, potentially skewing results.
Yes, the conclusions and interpretations are sound, logically derived from the presented data and analysis.
Yes, the references are properly cited, ensuring that all sources are appropriately credited and relevant literature is acknowledged.
Yes, this is a new and original contribution, offering fresh insights into the connection between oral health and stroke risk factors.
It is advised that this work be divided into two parts to facilitate easier reading. Percent match: 40% iThenticate report
Author Response
The authors very much appreciated the constructive comments on this manuscript by the reviewer. The comments have been very thorough and useful in improving the manuscript.
Comment 1: The summary is clear but could be more concise by eliminating repetitions and simplifying the objective, which is the relationship between the oral microbiome and stroke
Response 1: We have made changes to the abstract and introduction.
Comment 2: Other confounding factors, such as socioeconomic status and access to healthcare, are not deeply explored, potentially skewing results.
Response 2: Yes, you are right, such information would be appropriate. We have made corresponding changes to the manuscript.
Comment 3: It is advised that this work be divided into two parts to facilitate easier reading. Percent match: 40% iThenticate report
Response 3: Unfortunately, we do not have the opportunity to use the iThenticate service, but we have worked to improve the originality of the manuscript.
We also re-structured the manuscript and made a number of additional changes.
Reviewer 4 Report
Comments and Suggestions for Authors
This submission is registered within the Oral Health - Systematic Health theme. The goal is to produce a present condition report on the relationship between oral microbiota and stroke.
I have my doubts about the acquisition of new information. More to the point, the majority of the proposed text does not meet the project's goal. I'd suggest restructuring it and removing anything that doesn't add value (off-topic).
We have to wait for section 3 “Oral disease and stroke” to get to the heart of the matter. Section 6 is relevant. Isn't that the goal of this research review? As advocated in the conclusion (647-651)
A few comments:
The introduction can be simplified and made more effective. Example: Lines 72-76. Note that talking about (developing) biomarkers is an interesting idea. Just as the present classification of periodontal diseases includes systemic diseases.
What is the logic of the chapter on pregnant women (&2)? Line 101-173. I don't get it.
Line 175-183?
Line 184-186: What references do you have?
Line 187-191 is based on a 1982 meta-analysis (69). Are you sure of your reference?
Line 262-263; 264-266: No reference.
Line 359-380: What's the connection with the subject?
Line 401-419: You focus on intestinal microbiota? why?
You hardly ever mention the impact of adequate hygiene on periodontal health. And what about associated risk factor like tobacco, alcool, etc..
For a submission that targets “Microorganims”, I find that the space given to oral microbiota is limited. The focus is on periodontal disease, with all the biases, confounding and heterogeneity that this pathology represents in clinical research surveys.
You don't include the latest research on the interdental microbiota of adolescents and young adults. This is an important area to include in a predictive and preventive approach. As is the concept of para-inflammation.
To include the link between periodontal disease and stroke is a bit simplistic. We're still operating on the logic that “I have periodontal disease”, so my risk is impacted. Which is logical. But what happens between the ages of 18 and 35 when the periodontium is mostly healthy (bleeding, gingivitis, inflammation, etc.)?
Are you sure you want to keep thinking about caries and cancer?
Perhaps you could highlight the strengths and/or weaknesses of current knowledge?
And if there are any research opportunities?
Author Response
The authors very much appreciated the constructive comments on this manuscript by the reviewer. The comments have been very thorough and useful in improving the manuscript.
Comment 1: The introduction can be simplified and made more effective. Example: Lines 72-76. Note that talking about (developing) biomarkers is an interesting idea. Just as the present classification of periodontal diseases includes systemic diseases.
Response 1: The introduction has been revised in accordance with the feedback provided.
Comment 2: What is the logic of the chapter on pregnant women (&2)? Line 101-173. I don't get it. Line 175-183?
Response 2: We have removed this section. The part describing in vivo studies investigating the effects of oral bacteria on offspring brain health has been moved to Section 5.
Comment 3: Line 184-186: What references do you have?
Response 3: Corrected. We have made the appropriate changes to the manuscript.
Comment 4: Line 187-191 is based on a 1982 meta-analysis (69). Are you sure of your reference?
Response 4: This meta-analysis was conducted in 2012 and includes an analysis of several studies.
Comment 5: Line 262-263; 264-266: No reference.
Response 5: This paragraph does not provide any new data, but only summarises the information provided in the section
Comment 6:Line 359-380: What's the connection with the subject? Line 401-419: You focus on intestinal microbiota? Why?
Response 6: One of the mechanisms by which the oral microbiome exerts a systemic effect may be through the transfer of oral microorganisms to the gut and subsequent changes in the composition of the gut microbiome.
Comment 7: You hardly ever mention the impact of adequate hygiene on periodontal health. And what about associated risk factor like tobacco, alcool, etc..
Response 8: Corrected. We have made the appropriate changes to the manuscript
Comment 9: For a submission that targets “Microorganims”, I find that the space given to oral microbiota is limited. The focus is on periodontal disease, with all the biases, confounding and heterogeneity that this pathology represents in clinical research surveys.
Response 9: Yes, you are right. Unfortunately, there are not many studies investigating the relationship between the oral microbiome and stroke, and they have limited sample sizes. While the relationship between oral diseases and stroke is much better understood, with several large studies published in recent years, diseases such as periodontitis and dental caries are directly related to the activity of microorganisms.
Comment 10: You don't include the latest research on the interdental microbiota of adolescents and young adults. This is an important area to include in a predictive and preventive approach. As is the concept of para-inflammation.
Response 10: We added several studies including young adults, primarily with cryptogenic stroke.
Comment 11: To include the link between periodontal disease and stroke is a bit simplistic. We're still operating on the logic that “I have periodontal disease”, so my risk is impacted. Which is logical. But what happens between the ages of 18 and 35 when the periodontium is mostly healthy (bleeding, gingivitis, inflammation, etc.)?
Response 11: This is a very good question, but unfortunately we did not find any studies that looked at bleeding gums/gingivitis and stroke in young adults.
Comment 11: Are you sure you want to keep thinking about caries and cancer?
Response 11: The etiology of dental caries is linked to the oral microbiome and there are several large studies evaluating the relationship between dental caries and stroke. Yes, you are right. We have removed information about oral cancer.
Comment 12: Perhaps you could highlight the strengths and/or weaknesses of current knowledge?
Response 12: Information about the limitations of existing studies has been added.
Comment 13: And if there are any research opportunities?
Response 13: We have added information about possible directions for future research.
We also re-structured the manuscript and made a number of additional changes.